# Two Majority Voting Classifiers Applied to Heart Disease Prediction

**Talha Karadeniz [1,\*], Hadi Hakan Maraş [2], Gül Tokdemir [3] and Halit Ergezer [4]**

[1]  KUTTAM, School of Medicine, Koc University, 34450 Istanbul, Turkey
[2]  Department of Computer Programming, Vocational School, Cankaya University, 06790 Ankara, Turkey
[3]  Department of Computer Engineering, Faculty of Engineering, Cankaya University, 06790 Ankara, Turkey
[3]  Department of Mechatronics Engineering, Faculty of Engineering, Cankaya University, 06790 Ankara, Turkey
[\*]  Correspondence: tkaradeniz@ku.edu.tr

**Abstract:** Two novel methods for heart disease prediction, which use the kurtosis of the features and the Maxwell–Boltzmann distribution, are presented. A Majority Voting approach is applied, and two base classifiers are derived through statistical weight calculation. First, exploitation of attribute kurtosis and attribute Kolmogorov–Smirnov test (KS test) result is done by plugging the base categorizer into a Bagging Classifier. Second, fitting Maxwell random variables to the components and summating KS statistics are used for weight assignment. We have compared state-of-the-art methods to the proposed classifiers and reported the results. According to the findings, our Gaussian distribution and kurtosis-based Majority Voting Bagging Classifier (GKMVB) and Maxwell Distribution-based Majority Voting Bagging Classifier (MKMVB) outperform SVM, ANN, and Naive Bayes algorithms. In this context, which also indicates, especially when we consider that the KS test and kurtosis hack is intuitive, that the proposed routine is promising. Following the state-of-the-art, the experiments were conducted on two well-known datasets of Heart Disease Prediction, namely Statlog, and Spectf. A comparison of Optimized Precision is made to prove the effectiveness of the methods: the newly proposed methods attained 85.6 and 81.0 for Statlog and Spectf, respectively (while the state of the heart attained 83.5 and 71.6, respectively). We claim that the Majority Voting family of classifiers is still open to new developments through appropriate weight assignment. This claim is obvious, especially when its simple structure is fused with the Ensemble Methods' generalization ability and success.

**Keywords:** majority voting classifier; kurtosis; Gaussian distribution; Bagging Classifier; Ensemble Methods; heart disease prediction





## 1. Introduction

Given a dataset, Machine Learning is the craft of finding computational models from the collection of general observations. Medical Data Mining is a branch of Machine Learning which deals with healthcare data. After carefully modeling the input dataset, heart disease prediction automatically labels a fixed set of attributes related to heart disease.

The primary motivation behind the present paper was to develop new classifiers and tests in this critical field of medical data mining. Since human health is precious and applying technology in such a field is considerable, we have striven to improve the results.

Introducing new classifiers is not a common practice in this domain, and our methods can be seen as the result of an effort to fill this gap. Although the template of the proposed methods, the base estimators built upon majority voting schemes, is well known, introducing statistical weight assignment calculations derived via kurtosis or the Kolmogorov–Smirnov statistic, and additionally, plugging a specific density estimation method into the majority voting scheme contributes to the literature.

## 2. Related Work

Several studies have applied classification algorithms to the prediction of heart disease. Among them, [1] proposed a 'multi-criteria weighted vote-based classifier ensemble' to predict heart disease, while ref. [2] introduced a Chaos Firefly and Rough set-based feature reduction [3] (CFARS-AR) followed by a Fuzzy-Logic (FL) classifier [4] (CFARS-AR FL). Ref. [5] built a clinical decision support system through vote-based classifier ensembles, followed by an ensemble scheme again [6] for heart disease prediction. Another successful application of ensemble learning can be found in [7], where the authors reported a high accuracy for Random Forest.

Durairaj et al. [8] chose a Multilayer Perceptron model to make tests on the Cleveland dataset. Saqlain et al. [9] analyzed the Fisher score and Matthews correlation to apply feature selection before employing Support Vector Machines (SVM) for the classification. Cabral et al. [10] introduced a 'Feature Boundaries Detection' method for one-class classification and conducted experiments on the Spectf, Statlog, and Heart Murmur datasets.

Similar to the works of Bashir et al. [5,6], Vasudev et al. Although these are not 'pure ensemble' algorithms—such as Random Forests or Gradient Boosting Trees–they are ensemble methods and prove the correctness of a claim which can be derived [11]: that is, biomedical data is best analyzed via Ensemble Methods, even though Long et al. [2], which has been a guide for us, neither proposes an Ensemble Method nor carries out a comparison with Ensemble Methods. Instead, researchers in [2] construct a Fuzzy-Logic Classifier supported by Chaos Firefly and Rough Set Feature Reduction; we favor this hypothesis of Ensemble Methods by introducing a Bagging Classifier of Majority Voting variant weak categorizers.

In [12], Raghavendra et al. worked on a 'hybrid technique' consisting of Logistic Regression and ANN, reporting a cross-validation score for the Spectf dataset. Fitriyani et al. [13] studied the effectiveness of Extreme Gradient Boosting [14] when it is backed with DBSCAN outlier removal [15], and SMOTE-ENN balancing [16]. They report a high cross-validation score of 95.90 on the Statlog dataset. This also demonstrates the success of Ensemble Methods on medical datasets. Still, we have instead conducted our experiments according to the train–test split route noted in [2], for it belongs to one of most of the reviews and research on heart disease prediction up to date.

The proposed Majority Voting classifier resembles Naive Bayes (NB) ones since it processes the features individually (the difference is that in our approach, the feature classifications are combined through kurtosis + KS test or only KS test weighting). So, we have made an introductory survey of these classifiers. Despite its simplicity, NB Classifier [17] is still used in the literature: from intrusion, detection [18] to the preservation of privacy [19], from fake news detection [20] to emotion recognition [21], it has a wide range of applications. Variants of Naive Bayes are also effective. For example, a compression-based averaging scheme is proposed in [22,23]. Naive Bayes has been extended to a self-consistent variant to detect masquerades. [24] presents a locally weighted version of Naive Bayes to relax the independence assumption.

Kurtosis and Naive Bayes classifiers can be seen in the same context [25], restricted to Independent Component Analysis (ICA) dimension reduction. An example can also be seen in [26] where a 'quantile measure in ICA' is suggested. It is also a candidate for visual or generic feature extraction [27–30]. The literature on classifiers is still open to development. In addition, there are also works, such as [31], discovering the interpretation of state-of-the-art methods and adapting the classical methods to new systems [32]. To our knowledge, kurtosis has not previously been directly integrated as a weight calculation method in the Majority Voting scheme of Ensemble Methods. Hence this study aims to introduce an algorithm that integrates Majority Voting based Bagging classification with kurtosis statistical measure applied for weighting the features.

The latest work on heart disease prediction includes the Hybrid Random Forest Linear Model (HRLM) of [33], where the authors investigate the separate capabilities of each classifier before building their algorithm. Another hybrid method can be seen in [34], in

which Decision Tree (DT) and Random Forest (RF) are combined to predict heart disease. Results indicate that the combined method is much more powerful at classifying the data (Cleveland).

Ref. [35] compared *k*-Nearest Neighbors (*k*NN), DT, NB, and RF on Cleveland data and reported that the *k*NN approach was the most accurate one on a train–test split framework. Ref. [36] proposed a feature extraction-fusion approach backed with an ensemble deep learning method and presented an ontology-based system. Ref. [37] applied deep convolutional neural networks to tabular data for heart disease prediction, yielding high accuracy rates. This can be considered interesting because convolutional nets are not directly applicable to tabular data since an encoding-like pre-processing is generally needed before the convolutional feature extraction [38]. There are also generic works such as [39] where the capabilities of different optimizers for Deep Learning are researched.

On the other hand, to our knowledge, the Maxwell–Boltzmann distribution has never been used in the context of a general machine learning classification, which is the novel part of our work, at least in the sense that it is an integration of this probability density function (pdf) with a majority voting classifier scheme for the first time.

An assessment of majority vote point classifiers can be seen in [40], where the VC dimensions of majority vote point classifiers are explored theoretically and empirically. Under the guidance of [41,42], wherein the first one, the power of combining classifiers is demonstrated and in the second one, it has been shown that the sum rule yields the best classification results, we present our method: at the first level, a majority vote sum rule glued to statistical feature analysis is implemented, and at the second level, bagging of the base majority voting classifiers is employed to avoid overfitting.

## 3. Methodology

### 3.1. Dataset

Two datasets are used in the experiments to test the proposed prediction algorithms. The first one is UCI Spectf, where 'Single Proton Emission Computed Tomography' (SPECT) image features are stored. There are 44 integer features in the data, which are either stress or rest ROI counts between 0 and 100. The sample size is 267.

The second dataset is UCI Statlog, where the features are:

- age
- sex
- type of chest pain
- resting blood pressure
- serum cholesterol
- fasting blood sugar
- resting electrocardiography results
- maximum heart rate achieved
- exercise-induced angina
- old peak
- the slope of the peak exercise ST segment
- number of major vessels (0–3) colored by fluoroscopy
- thal

Statlog has a total of 270 observations, each having 13 features. A summary of datasets can be seen from Table 1.

**Table 1.** Dataset Properties.

| Name | Number of Features | Number of Samples | Source |
|---|---|---|---|
| Spectf | 44 | 267 | UCI |
| Statlog | 13 | 270 | UCI |

### 3.2. Model

In the data pre-processing stage, we have applied a Quantile Transformer [43] and a Robust Scaler [44], where the Interquartile Range (IQR) of the data is considered. Then, we have selected important features via Cross-Validated Recursive Feature Elimination (RFECV) [45], which is a cross-validated variant of [46]. An SVM estimator with a linear kernel is chosen to calculate the importance of each selected feature. Afterward, a Bagging Classifier which has a Logistic Regression [47] base estimator is stacked [48] with a Bagging Classifier which has a custom written base estimator. Our final estimator (i.e., 'meta-classifier') is also a Logistic Regression instance, combining the outputs of the bagged Logistic Regression and the bagged Majority Vote. The overall model can be seen in Figure 1.

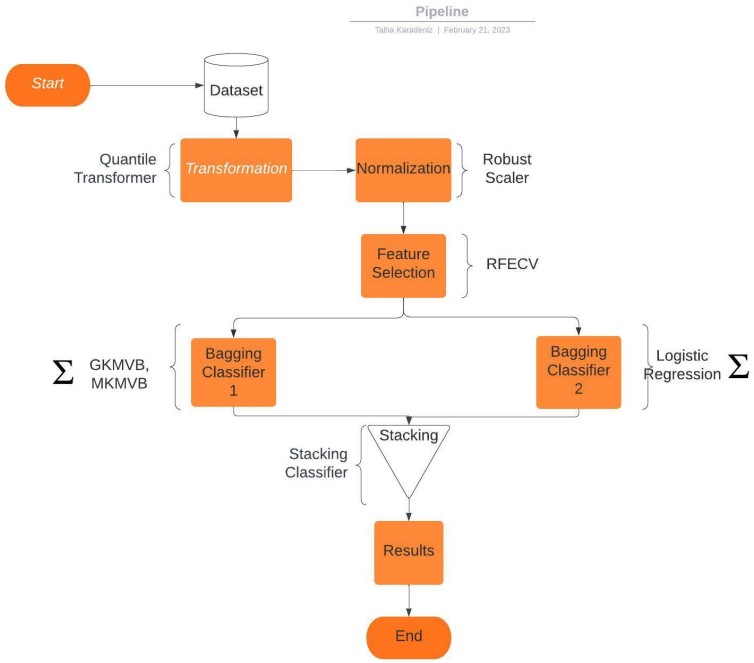

**Figure 1.** Prediction Model.

Our contribution relies on employing a majority voting scheme through attribute weighting calculations. For the first variant, $\exp(-|\kappa|)$, where $\kappa$ is the kurtosis, both it and the KS test results are plugged into the classifier by adding their values to the overall sum associated with the class label. The Maxwell–Boltzmann distribution is fitted to each attribute individually in the second variant. The KS test results of the fit are added to the overall sum of the winner class for the considered attribute.

The proposed Gaussian Distribution and Kurtosis-based Majority Voting Base Classifier Algorithm (GKMVB) is employed by a Bagging Classifier for heart disease prediction (Figure 2). The method is a binary classification scheme since Heart Disease Prediction has two classes. Details of the algorithms can be seen from Algorithm 1.

The base estimator comprises two functions: FIT() & PREDICT(). The FIT() function is used during training to calculate the statistical measures. In particular, the class-based mean, variance, and kurtosis values are calculated for each feature $i$. In the prediction phase, PREDICT() is employed, where we use the means and variances to calculate the Gaussian probability densities of the attributes, and majority voting is applied to determine the winning class. Additionally, we add the kurtosis and the KS statistic of the component to the overall value of the class vote.

The pseudo-code of the proposed algorithm is as follows:

---

**Algorithm 1** Proposed Base Estimator Method I: Gaussian Distribution based Majority Voting Classifier

---

1: **procedure** FIT($X, y$)                                                                          ▷ Dataset, class labels
2:     $means\_0 \leftarrow calc\_means(X, y, 0)$                                   ▷ Means of features for class 0
3:     $means\_1 \leftarrow calc\_means(X, y, 1)$                                   ▷ Means of features for class 1
4:     $vars\_0 \leftarrow calc\_variances(X, y, 0)$                              ▷ Variances of features for class 0
5:     $vars\_1 \leftarrow calc\_variances(X, y, 1)$                              ▷ Variances of features for class 1
6:     $kurtosis \leftarrow calc\_kurtosis(X)$                                          ▷ Kurtosis of each feature
7:     $ks \leftarrow ks\_test(X)$                                                       ▷ KS test result of each feature
8: **return** ($means\_0, means\_1, vars\_0, vars\_1, kurtosis, ks$)
9: **end procedure**

   **procedure** PREDICT($x, c\_0, c\_1, means\_0, means\_1, vars\_0, vars\_1, kurtosis, ks$)          ▷
   $means\_0, means\_1, vars\_0, vars\_1, kurtosis$, and $ks$ are results of the FIT procedure. $c\_0$
   and $c\_1$ are method parameters.
2:     $s\_0 \leftarrow 0$                                                                         ▷ Initialize votes for class 0.
       $s\_1 \leftarrow 0$                                                                         ▷ Initialize votes for class 1.
4:     $i \leftarrow 0$
       $D \leftarrow dim(x)$                                                                            ▷ Number of dimensions
6:     **while** $i < D$ **do**
           $val\_0 \leftarrow calc\_dens(x[i], means\_0[i], vars\_0[i])$          ▷ Given mean and variance,
       calculate density according to Equation (2)
8:         $val\_1 \leftarrow calc\_dens(x[i], means\_1[i], vars\_1[i])$                       ▷ Repeat for class 1
           **if** $val\_0 > val\_1$ **then**                ▷ Feature class based probabilities are compared
10:            $s\_0 \leftarrow c\_0 + exp(-|kurtosis[i]|) + ks[i]$          ▷ Kurtosis and KS-statistic added
           **else**
12:            $s\_1 \leftarrow c\_1 + exp(-|kurtosis[i]|) + ks[i]$
           **end if**
14:        $i \leftarrow i + 1$
       **end while**
16:    $y \leftarrow 0$                                                                                        ▷ Class of $x$
       **if** $s\_0 > s\_1$ **then**
18:        $y \leftarrow 0$                                                                                  ▷ Class of $x$ to 0.
       **else**
20:        $y \leftarrow 1$                                                                                  ▷ Class of $x$ to 1.
       **end if**
22: **return** $y$
   **end procedure**

---

For GKMVB, $X$ refers to the overall training data, whereas $x$ refers to an observation from the test set. PREDICT() is supposed to be executed for each observation in the test dataset. *val_0* and *val_1* are the Gaussian-based probability densities of the sample vector $x$, given class 0 and 1, respectively. $s_0$ and $s_1$ stand for the majority vote sums for each class. $exp(-|kurtosis[i]|)$ hack is used to obtain a scalar inversely proportional to the kurtosis of feature $i$ ($|kurtosis[i]|$). $ks[i]$ is directly used since its value is between 0 and 1.

$c_0$ and $c_1$ are set according to the class priorities, which are $c_0 = 2$ and $c_1 = 1$ for the Spectf dataset and $c_0 = c_1 = 1$ for for the Statlog dataset.

In the MKMVB variant, a Maxwell–Boltzmann random variable is fitted to each attribute's class components; that is, a fit is calculated for the samples belonging to class 0, and a second fit is calculated for the remaining samples (the samples of class 1). For each attribute, these two fits are stored to form a feature-level decision function. During the prediction stage, these feature-level decision functions are fused to determine the final class label via the majority vote sum rule (in the sum, the KS test values and class weights are used).

Another point is that our density calculation is a bit modified version of the normal pdf. For a normal distribution of location $\mu$ and scale $\sigma$, the probability density function is [49]

$$f(x) = \frac{1}{\sqrt{2\pi}\sigma} \exp(\frac{-(x-\mu)^2}{2\sigma^2}) \tag{1}$$

We instead used

$$g(x) = \frac{1}{c\sigma^4} \exp(\frac{-(x-\mu)^2}{2\sigma^4}) \tag{2}$$

to emphasize the effect of the variance on the classification, and we have seen that this gives better results in combination with kurtosis and KS statistic weighting.

Despite the fact that there are robust estimation routines for kurtosis [50], we have used the sample kurtosis [51]

$$\kappa = \frac{\hat{\mu}_4}{\hat{\sigma}^2} - 3 \tag{3}$$

where $\hat{\mu}_4$ is the sample moment defined by

$$\hat{\mu}_r = n^{-1} \sum_{i=1}^{n} (x_i - \bar{x})^r \tag{4}$$

and $\hat{\sigma}^2$ is the sample variance.

Our second method, MKMVB (details can be seen from Algorithm 2), differs from GKMVB in that the probability density function is used as given in [52] $(x, \theta > 0)$

$$f(x) = \frac{4}{\sqrt{\pi}} \frac{1}{\theta^{3/2}} x^2 \exp(-x^2/\theta) \tag{5}$$

and its CDF is

$$F(x) = \frac{1}{\Gamma(3/2)} \Gamma(\frac{3}{2}, \frac{x^2}{\theta}) \tag{6}$$

where $\Gamma$ is the incomplete gamma function:

$$\Gamma(a, x) = \int_0^x u^{a-1} e^{-u} \, du \tag{7}$$

On the other hand, for MKMVB, the weighting is done by summing the separate KS statistic measures on two random variable fits. The details of the MKMVB base estimator algorithm is as follows:

---

**Algorithm 2** Proposed Base Estimator Method II: Maxwell–Boltzmann Distribution based Majority Voting Classifier

---

    **procedure** FIT($X, y$)                                                      ▷ Dataset, class labels
        $rv\_0 \leftarrow fit\_maxwell\_rv(X, y, 0)$ ▷ Maxwell–Boltzmann random variables for class 0
        $rv\_1 \leftarrow fit\_maxwell\_rv(X, y, 1)$                    ▷ Random variables for class 1
        $ks \leftarrow ks(X, y, 0) + ks(X, y, 1)$             ▷ KS test result of each feature
    **return** ($rv\_0, rv\_1, ks$)
    **end procedure**
    **procedure** PREDICT($x, c\_0, c\_1, rv\_0, rv\_1, ks$)      ▷ $rv\_0, rv\_1$, and $ks$ are results of the FIT procedure. $c\_0$ and $c\_1$ are method parameters.
        $s\_0 \leftarrow 0$                                                     ▷ Initialize votes for class 0.
        $s\_1 \leftarrow 0$                                                     ▷ Initialize votes for class 1.
        $i \leftarrow 0$
        $D \leftarrow dim(x)$                                            ▷ Number of dimensions
        **while** $i < D$ **do**
            $val\_0 \leftarrow pdf(x[i], rv\_0[i])$ ▷ Given random variable, calculate probability density of $x[i]$.
            $val\_1 \leftarrow pdf(x[i], rv\_1[i])$                      ▷ Repeat for class 1
            **if** $val\_0 > val\_1$ **then**          ▷ Feature class based densities are compared
                $s\_0 \leftarrow c\_0 + ks[i]$                  ▷ KS-statistic added
            **else**
                $s\_1 \leftarrow c\_1 + ks[i]$
            **end if**
            $i \leftarrow i + 1$
        **end while**
        $y \leftarrow 0$                                                       ▷ Class of $x$
        **if** $s\_0 > s\_1$ **then**
            $y \leftarrow 0$                                        ▷ Class of $x$ to 0.
        **else**
            $y \leftarrow 1$                                         ▷ Class of $x$ to 1.
        **end if**
    **return** $y$
    **end procedure**

---

### 3.3. Training

To be consistent with the state-of-the-art methods, the steps given in [2] are followed: the first 80 instances of Spectf are reserved for training, and the remaining 187 instances are used for testing. The first 90 of Statlog are set to be the training set, and the remaining 180 observations are reserved for the testing set.

For Logistic Regression Bagging Classifier, the number of features ratio is 1.0 (i.e., all of the features are used), the number of estimators is 30, and the number of samples ratio is 0.5.

For GKMVB, the Bagging Classifier parameters are as follows: the number of features ratio is 0.5, the number of estimators is 20 and the number of samples ratio is 0.37 (MKMVB, our second method, uses the same Bagging Classification parameters).

There are two parameters of GKMVB and MKMVB: $c_0$ and $c_1$. $c_0$ is added to the sum associated with class 0 when the considered attribute's probability density function (i.e., the function $g$ in Equation (2) or $f$ in Equation (5)) for the class 0 is greater than that of class 1. $c_1$ is defined similarly. $c_0$ and $c_1$ of the GKMVB base classifier are set as $c_0 = 2$ and $c_1 = 1$ for the Spectf dataset and $c_0 = c_1 = 1$ for the Statlog dataset. On the other hand, the MKMVB base estimator parameters are set as $c_0 = 2.0$ and $c_1 = 1.0$. Figure 2 shows the general approach of the bagging classifier used in this study.

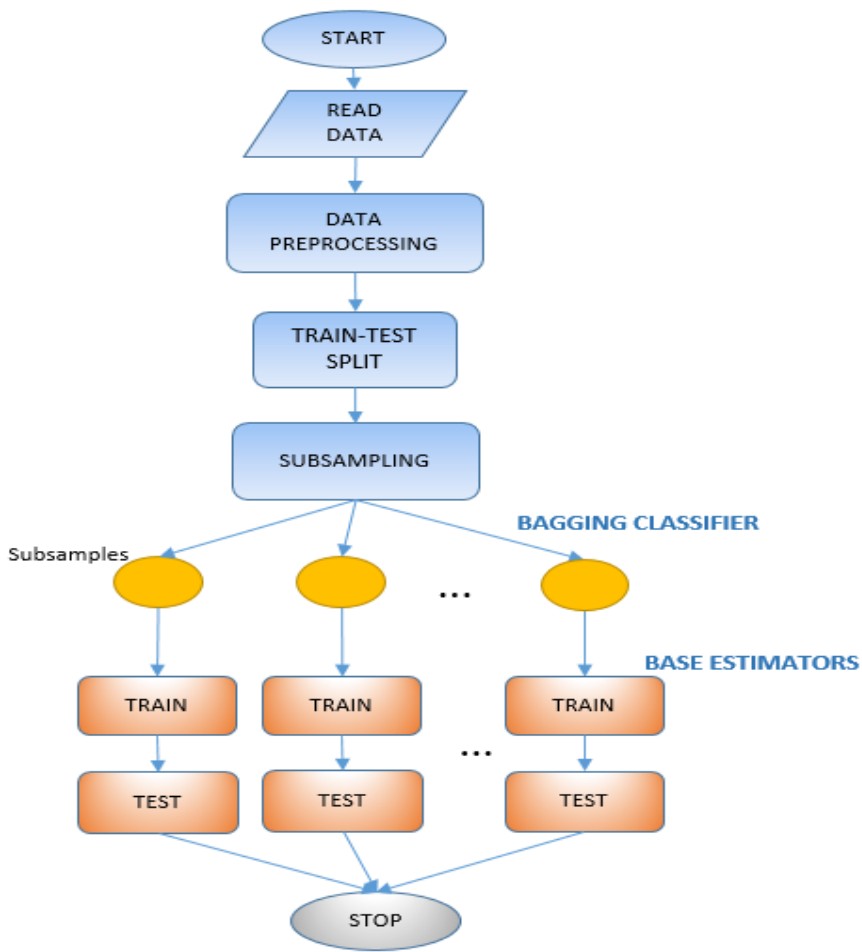

**Figure 2.** GKMVB & MKMVB Setup.

*3.4. Evaluation and Statistical Analysis*

For the performance comparisons, we have used four measures. The first is accuracy:

$$Accuracy = \frac{TP + TN}{TP + TN + FP + FN} \tag{8}$$

where *TP*, *TN*, *FP*, and *FN* are True Positives, True Negatives, False Positives, and False Negatives, respectively.

The second is the sensitivity (*Sn*);

$$Sn = \frac{TP}{TP + FN} \tag{9}$$

The third is the specificity (*Sp*);

$$Sp = \frac{TN}{TN + FP} \tag{10}$$

The last one is the Optimized Precision (OP) [53], which is a combination of the accuracy, the sensitivity, and the specificity:

$$OP = Accuracy - \frac{|Sn - Sp|}{(Sn + Sp)} \tag{11}$$

Using the *OP*, one can find the best candidate using all the measurements of accuracy, sensitivity, and specificity.

## 4. Results

To test the performance of the GKMVB algorithm, several experiments were conducted. Our experiments were run in the Google Colab [54] environment (Intel(R) Xeon(R) CPU @ 2.20GHz) with sklearn [55].

The OP scores can be seen in Tables 2 and 3, where CFARS-AR FL stands for the algorithm proposed in [2], and CFARS-AR NB, CFARS-AR SVM, CFARS-AR ANN stand for NB, SVM, and ANN backed with the feature selection process proposed in [2]. The accuracy, sensitivity, and specificity of CFARS-AR NB, CFARS-AR SVM, and CFARS-AR ANN are obtained from [2] (they have the default configurations of WEKA [56]).

In Tables 2 and 3, 'Proposed Method-I' and 'Proposed Method-II' stand for GKMVB and MKMVB, respectively.

We have also tested the scenarios where the preprocessing (Robust Scaling + Quantile Transformation + RFECV Feature Selection) remains the same while the base estimator is one of the state-of-the-art classifiers NB, SVM, ANN, or other ones such as DT, Perceptron, Passive Aggressive Classifier (PAC), Linear Discriminant Analysis (LDA) and Gaussian Process Classifier (GPC). The results are in Tables 4 and 5.

**Table 2.** Spectf Results.

| Method | Accuracy | Sensitivity | Specificity | OP |
|---|---|---|---|---|
| CFARS-AR NB | 79.7 | 100.0 | 0.0 | −20.0 |
| CFARS-AR SVM | 79.7 | 100.0 | 0.0 | −20.0 |
| CFARS-AR ANN | 77.0 | 89.3 | 28.9 | 25.9 |
| CFARS-AR FL | 87.2 | 94.2 | 68.9 | 71.6 |
| **Proposed Method-I** | 87.6 | 73.3 | 88.9 | **78.0** |
| **Proposed Method-II** | 83.1 | 80.0 | 83.4 | **81.0** |

**Table 3.** Statlog Results.

| Method | Accuracy | Sensitivity | Specificity | OP |
|---|---|---|---|---|
| CFARS-AR NB | 85.2 | 82.6 | 87.1 | 82.5 |
| CFARS-AR SVM | 81.5 | 82.6 | 80.6 | 80.2 |
| CFARS-AR ANN | 81.5 | 82.6 | 80.6 | 80.2 |
| CFARS-AR FL | 88.3 | 84.9 | 93.3 | 83.5 |
| **Proposed Method-I** | 88.3 | 91.7 | 84.3 | **84.1** |
| **Proposed Method-II** | 87.2 | 88.6 | 85.5 | **85.4** |

**Table 4.** Spectf Results with different base estimators.

| Base Estimator | Accuracy | Sensitivity | Specificity | OP |
|---|---|---|---|---|
| NB | 84.0 | 73.0 | 84.0 | 76.0 |
| SVM | 79.0 | 73.0 | 80.0 | 75.0 |
| ANN | 84.0 | 66.0 | 86.0 | 72.0 |
| DT | 86.5 | 73.3 | 87.7 | 77.5 |
| Perceptron | 83.7 | 60.0 | 85.8 | 65.9 |
| PAC | 78.6 | 93.8 | 74.6 | 73.6 |
| LDA | 84.8 | 73.3 | 85.8 | 76.9 |
| GPC | 85.3 | 53.3 | 88.3 | 60.6 |
| **Proposed Method-I** | 88.0 | 73.0 | 89.0 | **78.0** |
| **Proposed Method-II** | 83.0 | 80.0 | 83.0 | **81.0** |

**Table 5.** Statlog Results with different base estimators.

| Base Estimator | Accuracy | Sensitivity | Specificity | OP |
|---|---|---|---|---|
| NB | 87.0 | 96.0 | 76.0 | 75.0 |
| SVM | 88.0 | 92.0 | 82.0 | 81.0 |
| ANN | 85.0 | 94.0 | 76.0 | 75.0 |
| DT | 85.5 | 90.7 | 79.5 | 78.9 |
| Perceptron | 85.5 | 88.6 | 81.9 | 81.6 |
| PAC | 85.0 | 93.8 | 74.6 | 73.6 |
| LDA | 85.5 | 90.7 | 79.5 | 78.9 |
| GPC | 85.5 | 94.8 | 74.6 | 73.6 |
| **Proposed Method-I** | 88.0 | 92.0 | 84.0 | **84.0** |
| **Proposed Method-II** | 87.0 | 89.0 | 85.0 | **85.0** |

## 5. Discussion

The proposed Majority Voting Algorithms (GKMVB and MKMVB) have several advantages. The performance comparison given in Table 2 shows that the proposed methods outperform CFARS-AR SVM, CFARS-AR ANN, CFARS-AR NB, and CFARS-AR FL in terms of the optimized precision of the classification on both datasets. An additional point is that our method is more successful on unbalanced datasets with respect to the balanced dataset, which is quite rare in the medical field.

The success of MKMVB on Spectf can be explained roughly by the capability of the Maxwell–Boltzmann distribution to capture the spatial characteristics of the SPECT images, which needs further investigation. Moreover, the 'quasi'-density function given in Equation (2) also needs further investigation. From an empirical point of view, the logic behind the general separation capability of this function can be explained by its emphasis on the variance.

Another point that separates our work from the others is the maximal usage of ensemble learning: first, in the base estimator level where the majority voting is applied; Second, in the bagging phase, where the subsampling is applied; Lastly, in the stacking phase where the classifiers are fused. This three-fold ensemble structure makes the model robust.

One disadvantage of the proposed method is the dependence on the random nature of Bagging Classifiers. Although it almost took two or three trials to get the optimal random state, a Bagging Classifier having a higher average OP (and possibly accuracy) than CFARS-AR FL is, of course, a better option.

One could claim that the methods are too 'handcrafted' due to the statistical computations of the attributes, which can be seen as 'against the spirit of Machine Learning'. While this critique is partly true, we think that making a statistical analysis and grounding the work on the output of this analysis is suited to the framework of 'Statistical Learning Theory' as long as the analysis is automatic. Albeit there is a lack of a specific theoretical assessment, such as [40], we think that assisting the majority vote sum classifiers by characteristics of the distribution of the random variable does no harm. Moreover, since the two datasets have distinct feature characteristics, good optimized precision results imply the 'generic classifier' potential of the proposed methods.

For future work, we can note that a density estimation other than a Maxwell–Boltzmann or Gaussian, which would be more accurate and novel than the one at hand, can be developed. Regression variants can be plugged into a Logitboost framework, or more sophisticated new base classifiers can be designed. Classical probability distributions can be evaluated to find the most suitable one for a majority voting scheme. Normality tests other than kurtosis measures can be used or engineered. This work is modular in that all of its methods for 'density estimation', 'normality', and 'voting' can be replaced by more efficient ones.

**Author Contributions:** Conceptualization, H.H.M.; methodology, T.K.; software, T.K.; validation, H.E. and G.T.; formal analysis, H.H.M.; investigation, H.E.; resources, G.T.; data curation, T.K.; writing—original draft preparation, T.K.; writing—review and editing, G.T.; visualization, G.T.; supervision, H.E.; project administration, H.H.M. All authors have read and agreed to the published version of the manuscript.

**Funding:** This research received no external funding.

**Informed Consent Statement:** This article does not contain any studies with human participants or animals performed by any of the authors.

**Data Availability Statement:** Spectf and Statlog data can be retrieved from the UCI repository (accessed on 10 February 2023). https://archive.ics.uci.edu/ml/datasets/SPECTF+Heart, https://archive.ics.uci.edu/ml/datasets/statlog+(heart).

**Acknowledgments:** We would like to thank Yusuf Karacaören, Mehmet Fatih Karadeniz and Ahmet Serdar Karadeniz for their continuous support.

**Conflicts of Interest:** We certify that there is no actual or potential conflict of interest in relation to this article.

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
