# Peer review of "Two Majority Voting Classifiers Applied to Heart Disease Prediction"

_applsci, doi:10.3390/app13063767_

Round 1

Reviewer 1 Report

In this work, Kurtosis values of features and Maxwell-Boltzmann Distribution are considered in developing two novel methods for heart disease prediction. This work is meaningful. Some suggestions are as follows.

1. The structure of Figure 1 should be more compact and the font should be larger. Fig. 2 is unclear.

2. The motivation of the proposed method should be provided in more details.

3. More experiments should be done to support the conclusions.

Author Response

In this work, Kurtosis values of features and Maxwell-Boltzmann Distribution are considered in developing two novel methods for heart disease prediction. This work is meaningful. Some suggestions are as follows.

  1. The structure of Figure 1 should be more compact and the font should be larger. Fig. 2 is unclear.

Thank you for the comment. Accordingly, Figure 1 is redrawn and Figure 2 is modified so that it can be understood easily.

  1. The motivation for the proposed method should be provided in more details.

Thank you for the comment. We have added a motivation paragraph to introduction section as below:

“The main motivation behind this work was to develop new classifiers and to present these in the critical field of medical data mining.”

  1. More experiments should be done to support the conclusions.

Thank you for the comment. We have added new base estimator experiments and extended Table 4 and Table 5 accordingly. Newly tested base estimators are DT, Perceptron, Passive Aggressive Classifier, LDA and Gaussian Process Classifier.

You may access the modified paper from the address:

https://www.overleaf.com/read/zzbtqvyzgdyj

Reviewer 2 Report

The proposed methos used Voting approach and two base classifiers are derived through statistical weight calculation.  The hybrid model using Gaussian distribution and Kurtosis-based Majority Voting Bagging Classifier Gaussian distribution and Kurtosis-based Majority Voting Bagging Classifier (GKMVB) 15 and Maxwell Distribution-based Majority Voting Bagging Classifier (MKMVB) o 15 and Maxwell Distribution-based Majority Voting Bagging Classifier (MKMVB)  is developed.  They have produced the sufficient results to prove their point however few improvements could be taken place.

1- Introduction section -  Add motivation of the study, Discuss the research gap and also explain how those gap have been filled by proposed method.

2. Related work section is missing , so Related work need to be updated thoroughly, latest work must be included in the related work section. also the add the following work. Doi: 10.1155/2022/2789760 + Doi: 10.1007/978-981-16-3690-5_135

3. Why Hybrid model is used ? what are the superiority of this (GKMVB) + (MKMVB) model over traditional method.

4.  Conclusion should be improve. Discuss the future scope of this model.

5. Ensure that all the work have been cite in the text sequentially.

Author Response

The proposed methods used Voting approach and two base classifiers are derived through statistical weight calculation.  The hybrid model using Gaussian distribution and Kurtosis-based Majority Voting Bagging Classifier Gaussian distribution and Kurtosis-based Majority Voting Bagging Classifier (GKMVB) 15 and Maxwell Distribution-based Majority Voting Bagging Classifier (MKMVB) o 15 and Maxwell Distribution-based Majority Voting Bagging Classifier (MKMVB)  is developed.  They have produced the sufficient results to prove their point however few improvements could be taken place.

  1. Introduction section - Add motivation of the study, Discuss the research gap and also explain how those gap have been filled by proposed method.

Thank you for the comment. We have added a motivation paragraph (“The main motivation…”) and a paragraph about the research gap “Introducing new classifiers…” to the Introduction section.

  1. Related work section is missing , so Related work need to be updated thoroughly, latest work must be included in the related work section. also the add the following work. Doi: 10.1155/2022/2789760 + Doi: 10.1007/978-981-16-3690-5_135

Thank you for the comment. We have added a Related Work section by including two new paragraphs about the current literature; “Latest work on heart disease prediction…” and “[36] compared k-nearest…”.  Here the provided articles are cited, too. First at the end of the first paragraph of Related Work and second, at the end of the eighth paragraph of  Related Work.

  1. Why Hybrid model is used ? what are the superiority of this (GKMVB) + (MKMVB) model over traditional method.

Thank you for the comment. We have added the third paragraph of the Discussion section (“Another point which separates…”). This paragraph mentions the ‘maximal usage’ of ensemble learning throughout the method to improve accuracy.

  1. Conclusion should be improved. Discuss the future scope of this model.

Thank you for the comment. We have added a future work paragraph to the Discussion section (“For the future work, we can note that…”) and emphasized new ideas and new directions.

  1. Ensure that all the work have been cite in the text sequentially.

Thank you for the comment. Works are cited sequentially and checked.

You may access the modified paper from the address:

https://www.overleaf.com/read/zzbtqvyzgdyj

Reviewer 3 Report

1.     One of the contributions claimed by the authors is that the proposed new method. This is incorrect. Simply converting an old model is not the definition of new model. And the model description does not support this claim.

2.     The data description is confusing. And  is inconsistent with the tables.

3.     Figure 1 can be better used for illustration by adding the description of each term A part of the current introduction section contains the existing studies in the field. It is recommended to put them into a separate related work section.

4.     There are grammar/expression issues throughout the paper. A careful round of proofreading is needed.

Author Response

  1. One of the contributions claimed by the authors is that the proposed new method. This is incorrect. Simply converting an old model is not the definition of new model. And the model description does not support this claim.

Thank you for the comment. We have added a paragraph about the ‘novelty’ of the work; mentioning that it is not ‘completely new’ but its main point namely coding new base estimators for bagging classifiers deserves originality consideration.

  1. The data description is confusing. And is inconsistent with the tables.

Thank you for the comment. Statlog data description is itemized and made less confusing. Spectf data description is corrected: ‘continuous variables’ -> ‘integer features’.  Added a table about dataset properties (Table 1).

  1. Figure 1 can be better used for illustration by adding the description of each term A part of the current introduction section contains the existing studies in the field. It is recommended to put them into a separate related work section.

Thank you for the comment. Figure 1 is revised to be clearer and more visible (font size increased). Figure 2 renewed. Additionally, Related work section is added.

  1. There are grammar/expression issues throughout the paper. A careful round of proofreading is needed.

Thank you for the comment. Grammar is corrected by a native English speaker.

You can access the modified paper from the following address:

https://www.overleaf.com/read/zzbtqvyzgdyj

Round 2

Reviewer 2 Report

Authors has incorporated the comments in the revised version. I have no further comments.